# Demonstrating the Impact of Modelling Decisions

## Alan Lindsay, Ronald P. A. Petrick

Automated Planning Lab,
Department of Computer Science,
Heriot-Watt University, Scotland, UK
{alan.lindsay,r.petrick}@hw.ac.uk

## Abstract

The modelling problem involves making compromises between a variety of competing factors, including planning efficiency, plan language usefulness, and real-world optimisation goals. Optimal plans in the resulting model may appear suboptimal when executed in the world. As a consequence, a human observer might find it difficult to comprehend the apparent inefficient behaviours of the agent, which might impact on the human's trust of the agent. In this work we consider modelling decisions, such as abstractions, and their impact on the resulting plans. Our aim is to build a general approach that can assist a user to better understand both the implications of a modelling step and provide justification to support the modelling step. As a start, we have extended an off-the-shelf plan visualisation tool to provide plan failure visualisations, to demonstrate the impact of these modelling decisions to the user.

## Introduction

It has been quite common to consider the problem of explainability as a model based or model reconciliation problem (Chakraborti et al. 2017; Lindsay 2019; Eifler et al. 2020). Focusing on the agent's model can lead to the construction of many useful and relevant explanations, and assuming that the human's model is (partially) known in a similar representation has led to important results in XAIP. However, it is not yet clear whether this is sufficient in situations where the most relevant explanations lie outwith the scope of the models. In this work we consider an alternative to model reconciliation, where instead of expecting a human to 'fix' their model so that it better matches the robot's model, we instead consider informing the human about the agent's model, so that they might understand the reasons behind its decisions. In general the agent's model may simply be different, not necessarily better or worse in all cases. Moreover, many of the benefits of using AI approaches are not tied directly to better than human performance. For example, robustness, predictability, accountability can all be more important than reaching the goal in fewest actions in certain domains. However, providing the user with knowledge about the underlying agent model and the reasons for its selection should allow the user to use the agent better.

We consider the modelling problem, where the domain modeller makes a compromise between a variety of competing factors, including model comprehension, model conciseness, planning efficiency, plan language usefulness, real-world optimisation goals, modelling patterns. Many of these factors are unknown to the human user and might not correspond to the real-world optimisation criteria, or any other hidden criteria that the human user might care about. Moreover, decisions made during the modelling process will impact on the plans that will be generated. For example, when a problem model is abstracted, the cost of corresponding action sequences might be changed.

In this work, we consider the modelling process and how decisions that are made during modelling can impact on the plans generated by the resulting model. We identify the situations where the modelling decisions lead to suboptimal planning (in the context of some suitably rich model of the world). We consider a scenario where an agent presents a plan to a user and the user suggests an alternative plan, which they consider better. We consider the use of these plans to demonstrate how the agent's model has led to its plan, in order to assist the user in better understanding the agent's behaviour.

## Background

In this section we provide an overview of model representation and problem modelling.

**Problem Representation**  A classical planning problem can be defined as follows:

**Definition 1.** A Classical Planning Problem is a planning problem, $P = \langle F, A, I, G \rangle$, with fluents, $F$, actions, $A$, initial state, $I$, and goals, $G$. A solution (a plan) is a sequence of actions, $\pi = a_0, \ldots, a_n$, that transform the initial state, $I$, to a state, $s_n$, that satisfies the goals, $G \subseteq s_n$.

An action is defined by a precondition and an effect and is applicable in a state if its precondition is satisfied by the state. The set $S$ of states of a planning problem is the set of states that can be reached by applying any sequence of applicable actions to the initial state. We use $seq$, $seq_0$, $seq_1$ and similar to represent action sequences, e.g., $seq = a_0, \ldots, a_n$, and function $\texttt{cost}(seq)$ returns the cost of the sequence. The aim in classical planning is typically to find short plans (unit cost).

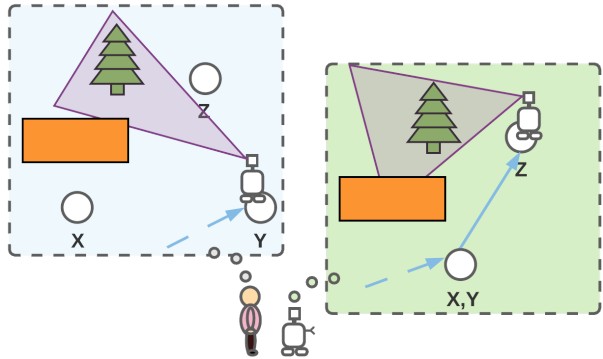

Figure 1: Two different models for the same problem. The agent's model (right) abstracts positions $X$ and $Y$ as a single position, from which the tree is not visible. In the user's model they are distinct and the tree is visible from $Y$.

Extensions to the classical planning problem include handling numeric constraints and optimisation (Fox and Long 2003), time (Fox and Long 2003), and uncertainty: in fully observable non-deterministic (FOND) planning problem (Muise, McIlraith, and Beck 2012) and partially observable planning problems (Bonet and Geffner 2011). And work in explainable planning, exists in many of these contexts, e.g., (Fox, Long, and Magazzeni 2017; Sreedharan et al. 2020; Lindsay et al. 2020; Carreno, Lindsay, and Petrick 2021; Porteous, Lindsay, and Charles 2021).

**Problem Modelling** We characterise the modelling process as a search through alternative models, with a set of features, $F_0, \ldots, F_n$ (e.g., computation time, number of actions), and a Boolean function, which determines the validity of the model. We make a simplifying assumption that the modelling process proceeds through a series of steps, from a rich model of the problem $M_n$ to a limited model of the problem $M_0$. For the purposes of this work, we assume two models: $M_i$ and $M_j$. The agent uses $M_i$ to plan and we assume that we can map the user's query plan onto a suitable plan in $M_j$. We also assume that we can map between structural elements (propositions and actions) in each model. For example, a scenario is presented in Figure 1, in which the agent (a robot) must take a photo of a tree. In the human model ($M_j$), positions $X$ and $Y$ are distinct and the human believes that the photo can be taken at $Y$. In the robot's model, $X$ and $Y$ are abstracted as position $X, Y$ and the robot must instead move further, to $Z$ to take the photo.

## Modelling Decisions

In this section we consider some individual simplifying steps that form part of this search. In each case we consider two models: $M_0$ and $M_1$, such that $M_1$ is a richer model and $M_0$ is the model after taking some simplifying step. We use $M_0$ and $M_1$ to represent any one of the series of steps made during the modelling process.

**An Abstraction Step** One modelling step is abstracting part of the model. For example, in our running example, $M_0$ has been abstracted from $M_1$ by a simple abstraction step that has merged locations $X$ and $Y$. As a result of an abstraction step, one state in the agent's model $s(M_0)$ represents a (potentially infinite) set of states $s_0(M_1), \ldots, s_n(M_1)$. We observe (unsurprisingly) that in making this step the model will not be as precise, which can lead to restrictions in the plans that the planner can discover. For example, in the running example, the agent's model combines $X$ and $Y$ and this abstraction means that it cannot take a picture from the resulting position $X, Y$.

In general, abstraction can cause variations in properties between the models, which can include optimality.

**Definition 2.** An optimality impairing modelling step $M_1$ to $M_0$ is a modelling step where relative optimality between action sequences is not preserved. More formally, there exists a pair of states, $s_b^1(M_1), s_e^1(M_1)$ and states that represent them in $M_0$, $s_b^0(M_0), s_e^0(M_0)$ and two action sequences, $seq_1^1(M_1), seq_2^1(M_1)$, which each transition from $s_b^1$ to $s_e^1$, and are represented in $M_0$ by $seq_1^0(M_0), seq_2^0(M_0)$, which transition from $s_b^0$ to $s_e^0$, and $\texttt{cost}(seq_1^1) < \texttt{cost}(seq_2^1)$ and $\texttt{cost}(seq_1^0) \geq \texttt{cost}(seq_2^0)$.

And abstraction can also impact on the valid sequences and solvability:

**Definition 3.** A sequence pruning modelling step $M_1$ to $M_0$ is a modelling step where abstracted states model fewer propositions. More formally, there exists an expression $\phi_0(M_0)$ and the equivalent expression $\phi_1(M_1)$, where in state $s(M_0)$, $\phi_0$ does not hold ($s \not\models \phi_0$), but where there is a state $s_j(M_1)$ in $M_1$, which is represented by $s(M_0)$, where $\phi_1$ holds ($s_j \models \phi_1$).

The use of abstractions is commonplace in planning, e.g., (Haslum et al. 2007; Newton et al. 2007; Gregory et al. 2011). An example has recently been implemented for abstracting from a hybrid continuous and discrete PDDL+ problem definition to a discretised PDDL2.2 model (Percassi, Scala, and Vallati 2021). As an example of a sequence pruning modelling step, discretising time can lead to certain action sequences being impossible due to the lost precision and this can also mean that some plans from $M_1$ are not considered plans in $M_0$.

**A Determinising Step** In FOND problems, each action has a set of possible outcomes (effects) and during execution one of these outcomes will occur (the planner cannot choose). One approach to these problems is to remodel the problem as a deterministic problem using a determinisation scheme (Muise, McIlraith, and Beck 2012). Two common schemes are the single actions determinisation (SOD), which selects one of the outcomes for each action (e.g., the one with most effects), and the all outcome determinisation (AOD), which creates an action for each of the outcomes. In the case of the SOD scheme, the determinised model might make the problem unsolvable. In the case of the SOD scheme, $M_0$ allows the individual outcomes to be selected. A promising action in $M_0$ might be linked with an alternative outcome in $M_1$, which actually leads to a dead-end. In

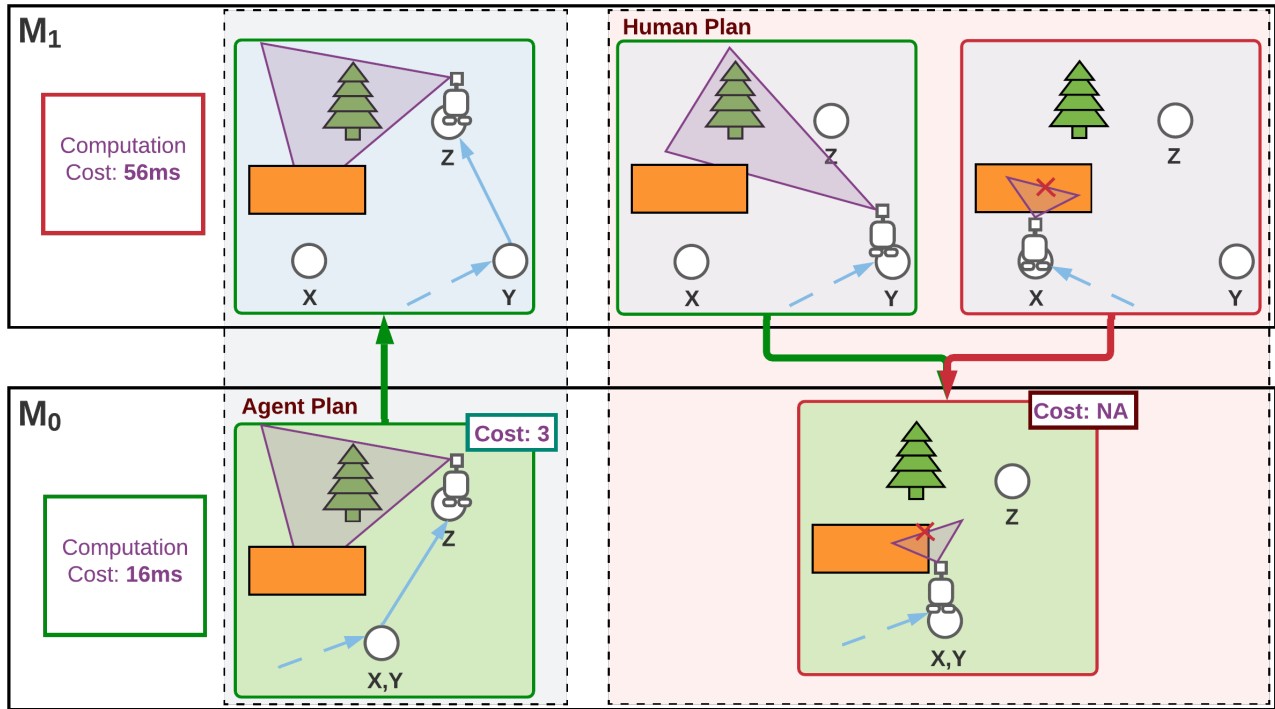

Figure 2: Aspects of the comparison between the agent's model ($M_0$) and a richer model ($M_1$). On the left is the associated computational cost of each model, providing justification for the modelling step. The mapping of the agent's plan is demonstrated, showing the plan in $M_1$, and justifying the agent's perception of the cost in $M_0$. On the right the mapping is used to demonstrate how the agent's abstract model leads to the user's plan, which is successful in $M_1$, being invalid in $M_0$.

contrast, an alternative plan might ensure success in the case of any outcome.

## Demonstrating the Impact of Modelling Decisions

Given the agent's proposed plan, $\pi_0(M_i)$ and the user's (mapped) counter proposal $\pi_1(M_j)$, we consider using the user's plan in order to demonstrate why the modelling steps made from $M_j$ to $M_i$ have led to the agent preferring plan $\pi_0$ over $\pi_1$. We have identified the following three requirements that we aim to satisfy in our work:

1. identify a set of modelling steps that are sufficient to explain the promotion of $\pi_0$ over $\pi_1$,

2. to explain and justify the identified modelling step(s),

3. demonstrate that the agent's plan is rational in the context of its model,

### Identifying Modelling Steps

Our intention is to use the comparison between the two plans to provide a context for identifying the relevant modelling steps. The first challenge will be to determine a model $M_j$, which is appropriate for mapping the user's alternative plan. We assume that $j > i$ (i.e., $M_j$ is richer) and that the equivalent of $\pi_0$ in $M_1$ costs more than $\pi_1$ (i.e., modelling steps might be attributable).

Given the sequence of steps, $j, \ldots, i$, and the associated models, each step, $k$ to $k-1$, can be labelled by examining the change in the model $M_k$ to $M_{k-1}$ (for the user's plan) and $M_{k-1}$ to $M_k$ (for the agent's plan). The labels are drawn from the potential impacts of each of the modelling steps (see Section 'Modelling Decisions'). For example, if the step has led to the user's plan being inconsistent, this step is associated with the label 'made inconsistent'. Similarly, in the case where a determinisation step has been made and a dead-end outcome has been added to the plan there will be a label 'added dead-end'. Notice, that although labels such as 'made suboptimal' would be useful, they would not be practical. Instead in other cases the labels note whether the abstraction directly impacts the structures (actions and propositions) that support the plan or not. We expect that the selection of appropriate modelling steps will be based on an ordering of the labels. For example, it is clear that a 'made inconsistent' label is sufficient.

In the remainder we assume that the steps to be visualised have been identified and we proceed with labelling the richer model $M_1$ and the limited model $M_0$.

### Explaining a Modelling Step

For each of the modelling steps that was indicated above, an explanation of the modelling step can be given to the user, with the aim of informing the user about the agent's model so that they are better able to judge the agent's abilities. Our aim is to create a visualisation and to use the comparison be-

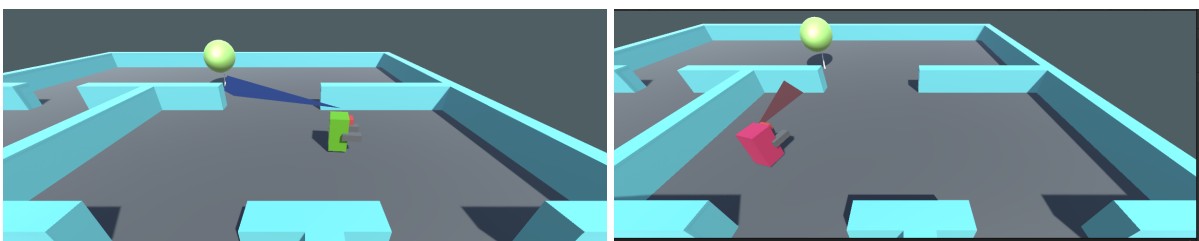

Figure 3: Two screenshots from PDSim illustrating the valuation of an expression, $\phi_1$ (is the tree visible?), in two states in $M_1$. On the left, $\phi_1$ holds: the robot is in a position where the tree is visible, whereas on the right, $\phi_1$ does not hold. The states both map to the same state ($s$) in $M_0$, and the tree is not visible in $s$.

tween the user and agent plans to support this explanation. Several of the components are illustrated in Figure 2, which is divided with $M_1$ shown at the top and $M_0$ at the bottom. On the left is a comparison of the features (more below) and next is the agent's plan, which can be visualised in both $M_0$ and $M_1$. In particular, the comparison between the plans in $M_0$ is important as this provides rational for the agent selecting the agent's plan ($\pi_0$) in place of the user's plan ($\pi_1$).

On the right $\pi_1$ is presented. In this example, we consider a sequence pruning (Definition 3) case of an abstraction modelling step. In this case, the impact of this abstraction can be visualised by first demonstrating plan failure of (the equivalent of) $\pi_1$ in $M_0$ and as a consequence establish that some state $s(M_0)$ does not model an expression $\phi_0(M_0)$ (the equivalent of $\phi_0$ is $\phi_1(M_1)$). Assuming that states $s_1(M_1), .., s_n(M_1)$ are represented by $s(M_0)$, then we can aim to discover two states ($s_i, s_j$), such that $s_i$ models $\phi_1$ ($s_i \models \phi_1$) and $s_j$ does not ($s_j \not\models \phi_1$). For example, two states for the running example are shown in top right of Figure 2. It might not be trivial to identify relevant (or any) elements from the states, $s_1, \ldots, s_n$, such as in the case of continuous variables. We can use the initial state of the problem to identify a relevant state. This is done by modifying the domain and problem: to find $s_j$ an action is added, which achieves the goal when i. the state is consistent with the mapping from state $s(M_0)$ to $M_1$ and ii. $s_j \not\models \phi_1$. In cases where a solution cannot be found we can attempt sampling methods to explore the mapped states in $M_1$.

**Visualising Plan Failure in PDSim**

In order to implement the steps indicated in Figure 2 we have began to implement our approach within PDSim (De Pellegrin and Petrick 2021), which is a plan visualisation tool implemented in the Unity game engine (https://unity.com/). The tool allows for objects to be associated with prefab objects and behaviours (e.g., animations) to be associated with action effects. We have introduced a proposition tester, which links state propositions with animations for positive and negative cases. We can therefore generate visualisations that demonstrate whether property $\phi_1(M_1)$ holds in state $s_i$ and state $s_j$.

In the case of our running example, we developed a simple behaviour to visualise whether the tree is visible from a specific location. First a ray is cast towards the tree. If it hits it first then a field of view is drawn in blue to indicate

the camera shot. If the ray hits another object first then the field of view is drawn in red and it stops at the first object. We used this approach on two states ($s_i$ and $s_j$) that were discovered using the above process. The visualisations are presented in Figure 3.

**Justifying the Modelling Step** It is also important that the user is able to understand the trade-offs that resulted in the modelling step being made. In this work we appeal to testing done during the viability study and the set of features associated with each model. These features provide the user with justification for each modelling decision, even if the features capture information that is irrelevant to them. For example, the conciseness of the model representation may be important for simplifying the description of interfaces with other parts of the system, but might be of no interest to the user. However, including all of the factors used for selecting the agent's model can aid the user by providing transparency. In Figure 2 (left) we have assumed the evaluation is based on a single feature: computation time, and that the acceptance function is based on a threshold value.

## Conclusion

In this paper we have considered problem modelling as a series of steps, such as abstraction or determinisation, that are made in order to optimise some factors. We consider how decisions made during this modelling process can have implications for the apparent quality of an agent's plans. This is contrary to a common assumption in XAIP, which is that the agent is using a model of the world to plan (i.e., optimality with respect to the agent's model is optimal in the world). We examine some specific modelling steps and the impact that these make on the resulting plans. We then present preliminary work that aims to demonstrate these impacts to the user through visualisations. Of course, the model is only part of the story and we are interested in how different types of explanations including modelling based explanations and planning explanations can be combined where appropriate. We are also interested in determining the best environment for communicating alternative models. There are now several off-the-shelf plan visualisers, e.g., (De Pellegrin and Petrick 2021; Kumar et al. 2022) and we aim to better understand the trade-offs between these alternative approaches.

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
