# OpenReview forum: "Demonstrating the Impact of Modelling Decisions"
_icaps-conference.org/ICAPS/2022/Workshop/XAIP — XAIP 2022_

### Official Review · Reviewer_56ot · 2022-04-16
**Interesting and possibly important problem of justifying model-abstraction decisions**

**Rating:** 6
**Confidence:** 4

**Review:**

Summary:
   This work considers how to explain model abstraction decisions to a human. This can be helpful  in order to explain why a human’s foil (counter proposal to the agent’s plan) is invalid or less preferable in the agent (robot’s) abstracted model. The assumption is that the human’s model is richer (model has more details) than the robot’s model in this particular problem setting. The authors define different abstraction step types that either affect the optimality of a plan, or removes plans (sequences) from the possible plans in a model. Model abstraction decisions maybe justified by the robot/agent along different criteria like computation time, or number of actions; criteria that are pertinent to the robot’s performance.
   I liked the attempt to characterize or distinguish different types of abstraction steps, with the objective of justifying decisions to a human agent later. I think more thought needs to be given about such justifications, but this is an interesting start.

Overall I think it's an interesting direction, and worth presenting at a workshop to share with the community and get feedback to improve it.

—--------------------------------------------------------------------------------------------------------------------

Questions and Concerns:

+ ) I am not sure what you want \phi_0 and \phi_1 to mean. I assume it is a logical expression over a particular set of propositions? … given that you used the “entails” symbol. Do you want to say that if a particular expression does not hold in an abstraction, then a plan that requires it to hold at some step will no longer be applicable ? i.e. a sequence pruning abstraction? Whether this is the case or not, the description could be made clearer with an example.

+ ) There was a lot of emphasis on plan visualization in PDsim, which I assume is for human subject studies, but no human studies were done. This space could have been better served by explaining your definitions with clean examples.

+ ) I am not sure about the value of describing FOND planning and then not using it as much in your paper. You could do the same with a simpler PDDL formulation and say an action was removed in the model abstraction because it’s benefit was only in a rare state that could be avoided, and the time cost of planning was higher. If FOND is somehow central to your work on model abstraction –which I can’t see how– I would make that connection clear.

+) Assuming the human’s model is necessarily richer is a very strong assumption. Can you make do with a weaker assumption? like richer w.r.t. only a particular plan, or say that the user is a domain expert who knows the ground truth model.

+) It is stated in the paper that “In this work we consider an alternative to model reconciliation, where instead of expecting a human to ‘fix’ their model so that it better matches the robot’s model, we instead consider informing the human about the agent’s model, so that they might understand the reasons behind its decisions”... this is still a type of model reconciliation by the way. The extra (and contribution) part is that you are adding justifications during the model reconciliation, and formalizing types of abstraction steps associated with these justifications.

You are still justifying the robot’s plan, or explaining why the human’s counter-plan is wrong, by explaining –“informing” as you say– about the model and/or performance differences. If your distinction is that you are not asking the human to update their model, I don’t think that is a helpful or important distinction to make, and is not a battle you need to fight. I would recommend selling the work as how to justify model abstraction steps, and build the work more in this direction. I think more thought on these justifications, and human studies need to be done.

When you give a justification, the human is going to update their model of the robot whether you ask them to or not. This is the model that they use to understand the robot behavior. See Chakraboty et al.’s work “Human-Aware Planning Revisited: A Tale of Three Models” . They clearly state that model reconciliation is either the human’s model of the robot (M^R_h) is brought closer to the model of the robot (M^R), or  robot’s model of the human(M^H_r) is brought closer to the or model of the human(M^H). What is described in the paper (correct me if I’m mistaken) is trying to explain and justify the differences between M^R_h and M^R.

+) In the paper, you assume that the human’s model ( M^R_h) is richer which I guess is justifiable if the human is a domain expert. Otherwise it is a very strong assumption. It might be worth making this clear

+) It is stated in the paper “Notice, that although labels such as ‘made suboptimal’ would be useful, they would not be practical”...what do you mean by “not be practical”. Please explain.

—--------------------------------------------------------------------------------------------------------------------

Suggestions:
Prefer writing a longer paper for future versionsl; 4 pages is needlessly short. You can make your definitions clearer with examples and add human subject studies .

Paper writing needs a lot of improvement. It was quite challenging to read. Currently, the last paragraph of the introduction describes what your approach is. It would help to follow this with another paragraph that informs the reader of the layout of the paper and how you will explain and justify your idea in the different sections.

I strongly suggest doing a human subject study, and perhaps use a non-motion based example to show your approach applies to more than just position abstraction.

—--------------------------------------------------------------------------------------------------------------------

Relevant Literature:

+) The following is a relevant paper that uses abstractions for explanations:
Hierarchical Expertise Level Modeling for User Specific Contrastive Explanations
https://www.ijcai.org/proceedings/2018/0671.pdf

+) Not all work assumes the robot’s model is the correct model of the environment (as you state in the conclusion).
Take a look at https://www.tandfonline.com/doi/full/10.1080/07370024.2020.1726751
this work does not assume that model of the robot is the true environment model.


—--------------------------------------------------------------------------------------------------------------------.

Typos:
In Introduction:

+) “lie outwith the scope”...do you mean lies outside the scope ?

+) “Moreover, many of the benefits of using AI approaches are not tied directly to better than human performance” … do you mean “not tied to directly to being better than human performance” ??

In Definition 2:

+) “And abstraction can also impact on the valid sequences and solvability”
reword to “Additionally, abstractions can also have an impact on the valid sequences and solvability of the problem”

In Explaining a modeling step:

+) “ the comparison between the plans in M0 is important as this provides rational for the agent selecting the actions”
…rational -> the rationale

In Visualizing Plan failure in PDSIM

+) “We have began” -> We have begun… I know, grammar is annoying :-)

---

### Official Review · Reviewer_QvDq · 2022-04-21
**review of "Demonstrating the Impact of Modelling Decisions"**

**Rating:** 7
**Confidence:** 4

**Review:**

This short paper presents a preliminary approach to explaining planning modeling decisions. This is in contrast to the prevalent approaches of explaining the agent's actions in the context of a given domain. As such, this approach aims to explain how modeling decisions affect the plan.

The idea is interesting, and to the best of my knowledge, it is novel.
The work seems still quite preliminary - it is not entirely clear to me whether there is an actual algorithm that identifies the modeling steps (it is written in a fairly general manner), what is the complexity, etc. There are also a few assumptions that are quite strong with respect to the available knowledge. And the approach has not yet been evaluated.

That said, I think this is OK for a workshop, and it will be interesting to discuss this work. Therefore, I recommend accepting this work.

---

### Meta-Review · Program_Chairs · 2022-04-30

**Recommendation:** Accept
**Confidence:** 5

**Metareview:**

The paper presents some ideas on explaining how the modeling decisions, such as abstractions, underlying a planning problem may affect its generated plans. Although in a preliminary stage, it is an interesting direction and would be a good addition to the workshop.

Nevertheless, we suggest the authors revise the paper with the reviewers’ comments. Particularly, there seems to be a confusion with the term “model reconciliation” and some of the assumptions about the human’s model (see reviewer’s 2 comment). It will benefit the paper greatly if these were addressed and ironed out.

We are looking forward to your presentation.

---

### Decision · Program_Chairs · 2022-04-30

Accept